# Crosstalk between Growth and Osmoregulation of GHRH-SST-GH-IGF Axis in Triploid Rainbow Trout (*Oncorhynchus mykiss*)

**DOI:** 10.3390/ijms23158691

**Published:** 2022-08-04

**Authors:** Kaiwen Xiang, Qian Yang, Mengqun Liu, Xiaodong Yang, Jifang Li, Zhishuai Hou, Haishen Wen

**Affiliations:** Key Laboratory of Mariculture, Ministry of Education (KLMME), Ocean University of China, Qingdao 266003, China

**Keywords:** triploid rainbow trout, *ghrh-gh-sst-igf* axis, osmoregulation

## Abstract

Smolting is an important development stage of salmonid, and an energy trade-off occurs between osmotic regulation and growth during smolting in rainbow trout (*Oncorhynchus mykiss*). Growth hormone releasing hormone, somatostatin, growth hormone and insulin-like growth factor (GHRH-SST-GH-IGF) axis exhibit pleiotropic effects in regulating growth and osmotic adaptation. Due to salmonid specific genome duplication, increased paralogs are identified in the *ghrh*-*sst-gh*-*igf* axis, however, their physiology in modulating osmoregulation has yet to be investigated. In this study, seven *sst* genes (*sst1a*, *sst1b*, *sst2*, *sst3a*, *sst3b*, *sst5*, *sst6*) were identified in trout. We further investigated the *ghrh*-*sst-gh*-*igf* axis of diploid and triploid trout in response to seawater challenge. Kidney *sst* (*sst1b*, *sst2*, *sst5*) and *sstr* (*sstr1b1*, *sstr5a*, *sstr5b*) expressions were changed (more than 2-fold increase (except for *sstr5a* with 1.99-fold increase) or less than 0.5-fold decrease) due to osmoregulation, suggesting a pleiotropic physiology of SSTs in modulating growth and smoltification. Triploid trout showed significantly down-regulated brain *sstr1b1* and *igfbp2a1* (*p* < 0.05), while diploid trout showed up-regulated brain *igfbp1a1* (~2.61-fold, *p* = 0.057) and *igfbp2a* subtypes (~1.38-fold, *p* < 0.05), suggesting triploid trout exhibited a better acclimation to the seawater environment. The triploid trout showed up-regulated kidney *igfbp5a* subtypes (~6.62 and 7.25-fold, *p* = 0.099 and 0.078) and significantly down-regulated *igfbp5b2* (~0.37-fold, *p* < 0.05), showing a conserved physiology of teleost IGFBP5a in regulating osmoregulation. The IGFBP6 subtypes are involved in energy and nutritional regulation. Distinctive *igfbp6* subtypes patterns (*p* < 0.05) potentially indicated trout triggered energy redistribution in brain and kidney during osmoregulatory regulation. In conclusion, we showed that the GHRH-SST-GH-IGF axis exhibited pleiotropic effects in regulating growth and osmoregulatory regulation during trout smolting, which might provide new insights into seawater aquaculture of salmonid species.

## 1. Introduction

Rainbow trout (*Oncorhynchus mykiss*) are an important salmonid fish cultured all over the world. Rainbow trout and steelhead are the same species (*Oncorhynchus mykiss*) with different lifestyles (National Wildlife Federation, NWF), rainbow trout are landlocked and spend their life mostly (or entirely) in freshwater; while steelhead is an anadromous species, and the juveniles start out in freshwater and then exhibit migration to marine environments [1,2]. Due to anthropogenic activities (with the resulting global climate change), freshwater is becoming increasingly precious, and a Nature paper reported that “80% of the world’s population is exposed to high levels of threat to water security” [3,4]. Global warming results in rising temperatures in ocean and freshwater ecosystems [5,6]. Two recent papers published in *Nature* series (2017 and 2021) demonstrated the global warming threats to global fishery biodiversity, especially the freshwater faunas [7,8]. Therefore, the two greatest challenges for salmonid culture are reduced freshwater resources (hatcheries) for juveniles and increased water temperature in summer. The Yellow Sea is a semi-closed part of the Pacific Ocean, which is near to the Chinese mainland. In the Yellow Sea, the Yellow Sea Cold Water Mass presents a cold-water mass in summer (<10 °C), which is excellent for summer salmonid culture [9].

Sexual maturation leads to significant reduction of somatic growth in fish. The triploid trout fry can be artificially manufactured by inhibiting the excretion of the second polar body after fertilization, such as hydrostatic treatment [10]. Compared to diploid species, triploid trout exhibited a better growth performance with increased market value due to infertility [11]. Moreover, farm diploid salmonid are known to escape, interbreed, and compete with wild salmonid and non-salmonid species, thus disturbing the aquatic ecosystem [12]. The escaped triploid individuals do not risk a great biological hazard to the natural environment and natural fish populations because of their sterility [13]. Given that salmonid species are not natural fish populations in the Yellow Sea Cold Water Mass, culturing infertile triploid trout is the most effective strategy for environmentally friendly aquaculture.

Salmonids exhibit significant appearance changes and physiological alterations during down-stream migration, which is called smolting [14]. For example, salmonids exhibit changes in morphology in response to increased salinity, including a more streamlined body, darker fin margins, loss of body spots, and silver throughout their bodies [15]. Fish osmoregulation is regulated by multiple endocrine regulators including the growth hormone (GH), and insulin-like growth factor (IGF) [15,16]. The GH-IGF axis is regulated by several endocrine regulators of growth hormone releasing hormone (GHRH), somatostatin (SST), GH receptor (GHR), IGF receptor (IGFR), and IGF binding proteins (IGFBPs) [17]. In addition to the canonical function of regulating growth, the GH-IGF axis also serves as a pleiotropic governor for immunomodulation and osmoregulatory regulation [18,19,20,21]. Previous studies have found that the injection of GH and IGF-1 can both improve salinity tolerance of juvenile Atlantic salmon (*Salmo salar*), and the effect of GH is better than IGF-1 [22]. Similarly, tilapia (*Oreochromis mossambicus*) stimulated with GH can enhance the salt resistance, and the Na^+^, K^+^-ATPase activity on gill [23].

IGFBPs not only regulate IGF signaling pathways and functions, but are also involved in diverse functions beyond IGF regulation [24]. Seawater exposure (30 ppt) results in significantly increased serum IGFBP1 subtypes rather than IGFBP-2B in Chinook salmon (*Oncorhynchus tshawytscha*) parr [25]. Microarray analysis showed that *igfbp1* had a strong relationship with high salinity adaptation in Coho Salmon (*Oncorhynchus kisutch*) [26]. Studies in Atlantic salmon showed expression of branchial *igfbp6b* subtypes coincidentally increased with smoltification while the *igfbp5* subtypes significantly decreased following seawater (35 ppt) exposure [27]. The following study further showed branchial exhibits *igfbp5b2* expression in parallel with Na^+^/K^+^-ATPase activity and transcriptional signature of ion transporters (channels) mRNA levels [28].

Smolting is an important process of development, physiology, and behavior, which enables wild trout migration to marine environments and farmed trout to be farmed in net cages in the sea [29,30]. The GHRH-SST-GH-IGF axis is involved in energy trade-off of multiple functions of growth, immunity, and osmoregulation [14,19,28]. The salmonid specific genome duplication results in increased paralogs in the *ghrh-sst-gh-igf* axis [31,32,33]. However, functional divergence of these novel paralogs in response to seawater exposure are still limited in diploid and triploid trout. To further understand the effect of the GHRH-SST-GH-IGF axis in seawater exposure between diploid and triploid trout and explore the novel regulatory mechanisms of trout osmoregulation and the candidates of genetic breeding for seawater trout culture, we investigated the *sst* gene repertoire, and then evaluated the physiology of the *ghrh-sst-gh-igf* axis in response to seawater exposure between diploid and triploid trout.

## 2. Results

### 2.1. Identification of Rainbow Trout sst Genes

Totally, seven *sst* genes (*sst1a*, *sst1b*, *sst2*, *sst3a*, *sst3b*, *sst5*, *sst6*) were identified in rainbow trout (Table 1), and these sst genes were localized in three different chromosomes of 9 (*sst2*, *sst5*, *sst6*), 15 (*sst1b*, *sst3a*), and 27 (*sst1a*, *sst3b*) (Figure 1A). The amino acids sequences of SST proteins ranged from 98 to 115 amino acids (Table 1). Multiple sequence alignments of trout SST with vertebrate orthologues are shown in Appendix A. The phylogenetic analysis showed trout *sst* could divide into six subgroups with two duplicates genes of *sst1* (*sst1a* and *sst1b*) and *sst3* (*sst3a* and *sst3b*) (Figure 1B). The *sst4* was absent in rainbow trout (Figure 1B). Exon-intron structure showed most of the *sst* genes exerted two exons while *sst3a* exhibited four exons (Figure 1C). Compared to two SST1 subtypes, a significant alteration of motif structures was observed between SST3a and SST3b, potentially leading to functional diversity (Figure 1C). Syntenic analysis showed *sst1* and *sst3* subtypes exerted conserved alignments with neighboring genes between rainbow trout and Atlantic salmon (Figure 1D). The *sst2*, *sst5*, *sst6* exhibited synteny homolog in teleosts (Figure 1E–G).

### 2.2. Basal Expression of ghrh-gh-sst-igf Axis in Triploid Trout

In brain, *ghrhrl1* and *ghrhrl2* exhibited high basal expression, while *ghrhrl1* exerted higher basal expression in kidney and liver (Figure 2A,G,M). Four *ghr* (*ghra1*, *ghra2*, *ghrb1* and *ghrb2*) subtypes showed higher basal expressions in brain, kidney, and liver, with the highest expression of *ghrb1*, *ghrb1*, and *ghra1* in brain, kidney, and liver, respectively (average expression count > 300, Figure 2B,H,N).

In brain, *sst1a*, *sst1b*, *sst2*, and *sst5* exhibited higher basal expressions, with average expression count great than 100 (Figure 2C). Compared to brain, kidney and liver exhibited relatively lower *ssts* expressions. The average expression levels of kidney *sst1b*, *sst2*, and *sst5* were ~5.9, 19.4, and 5.7, respectively (Figure 2I). Liver *sst1a*, *sst1b*, *sst3a*, and *sst3b* exhibited average expression levels of 11.2, 42.8, 6.5, and 37.7 respectively (Figure 2O). Brain exhibited higher basal expressions of *sstr1* and *sstr2* subtypes (except for *sstr2b1*) and *sstr3b* (Figure 2D). The basal expression of *sstr1a1*, *sstr2a2*, *sstr2b2* were relatively higher in kidney, and *sstr2a2*, *sstr2b1*, *sstr2b2*, and *sstr5b* showed higher basal expression in liver (Figure 2J,P).

Brain and kidney exhibited higher basal expressions of *igf1* subtypes (*igf1a1*, *igf1a2*, and *igf1b1/igf3*) and *igf2* with the receptors of *igf1ra1* and *igf1ra2* (Figure 2E,K). Liver exhibited extremely higher basal expressions of *igf1a1* and *igf1a2* with average expression count more than 750 (Figure 2Q). High basal expressions of *igf1b/igf3* and *igf2* were also observed in liver (Figure 2Q). Brain showed higher basal expressions of *igfbp2a1*, *igfbp2a2*, *igfbp4a*, *igfbp4b*, *igfbp5a2*, *igfbp5b1*, *igfbp5b2*, and *igfbp6b1* (average count > 50, Figure 2F). In kidney, high basal expression levels of *igfbp1a2*, *igfbp2a1*, *igfbp2a2*, *igfbp4a*, *igfbp4b*, *igfbp5b1*, *igfbp5b2*, and *igfbp6b1* were observed (average count > 50, Figure 2L). Liver showed extremely high basal expression levels of *igfbp1a1*, *igfbp1b1*, *igfbp1b2*, *igfbp2a1*, *igfbp2a2*, and *igfbp2b1* (average count > 1000, Figure 2R). High basal expressions of *igfbp1a2*, *igfbp4b*, *igfbp5b1*, and *igfbp6b1* were observed in liver (average count > 50, Figure 2R).

### 2.3. Comparison of Basal Expression of ghrh-gh-sst-igf Axis between Diploid and Triploid Trout

Compared to diploid trout, the top three upregulated genes in triploid brain were *sst3b*, *sstr3b*, and *sst5*, and the downregulated genes were *sstr5a*, *igfbp3b2*, and *igfbp6a1* (Table 2). In liver, triploid showed the top 3 upregulated genes of *igf1a2*, *igfbp4b*, and *sst3b*, and the top 3 downregulated genes of *igfbp6a1*, *igfbp3a2*, and *igfbp5a1* (Table 2). In kidney, the top 3 upregulated genes were *ghrh*, *igf1a2*, and *sstr5b*, and the top 3 downregulated genes were *ghrhrl2*, *igfbp3a2*, and *igfbp2a2* in triploid (Table 2).

### 2.4. Transcriptional Signature of Triploid Brain ghrh-gh-sst-igf Axis in Response to Seawater Challenge

Overall expression profiles of the brain *ghrh-gh-sst-igf* axis were shown via a heatmap (Figure 3A). The data were analyzed via unsupervised (PCA) and supervised (PLS-DA) methods for dimension reduction and discriminative variable selection. PCA showed clear discrimination between the DS and TS groups (Figure 3B), and *sst3b*, *sst6*, *sstr2a1*, *sstr3a*, *sstr5b*, *igfbp1b1*, *igfbp1b2*, *igfbp3b2*, *igfbp6a1*, and *igfbp6a2* served as key genes for PCA discrimination (Figure 3C). Clear discriminations were observed among groups of DS vs. TS and groups of TS vs. TF (Figure 3D), with key genes including *ghrh*, *sst6*, *sstr3a*, *sstr5b*, *igfbp1b1*, *igfbp3b2*, *igfbp6a1*, and *igfbp6a2* (Figure 3E). Compared to DS, the TS group exhibited increased expressions of *ghrh* and *igfbp6a2*, and decreased *sst6*, *sstr5b*, *igfbp1a1*, *igfbp1b2*, *igfbp2a1*, *igfbp2a2*, *igfbp3b2*, *igfbp6a1* (Figure 3F). Trout of the TS group showed increased expressions of *ghrh*, *sstr3a*, *igfbp1a2*, and decreased *sstr1b1*, *igfbp2a1*, and *igfbp6a1* compared to TF (Figure 3G).

### 2.5. Transcriptional Signature of Triploid Liver ghrh-gh-sst-igf Axis in Response to Seawater Challenge

The heatmap showed the overall transcriptional signatures of liver *ghrh-gh-sst-igf* axis (Figure 4A), and PCA and PLS-DA were used for dimension reduction. A clear discrimination between TF and TS was observed via PCA with key genes of *sstr1a1*, *sstr5a*, *igf1a1*, *igf1a2*, *igfbp1a1*, *igfbp1b1*, *igfbp5a1*, *igfbp6a2*, and *igfbp6b2* (Figure 4B,C). The supervised dimension reduction of PLS-DA exhibited distinct transcription signatures of liver *ghrh-gh-sst-igf* axis between DS and TS (Figure 4D). We observed that *sstr2a1*, *sstr5a*, *igf1a1*, *igf1a2*, *igf1ra1*, *igfbp1b1*, *igfbp5a1*, and *igfbp6a2* resulted in distinctly transcriptional discriminations of the *ghrh-gh-sst-igf* axis between DS and TS (Figure 4E). Compared to DS, the TS group exhibited increased expressions of *igf1ra1* and decreased *igf2*, *igfbp5a1*, *igfbp5a2*, and *igfbp6b2* (Figure 4F). The TS group exhibited increased expressions of *igf1ra1* and reduced *igf1a1*, *igf1a2*, *igf1b/igf3*, *igfbp4b*, *igfbp5a1*, *igfbp5a2*, and *igfbp6b2* (Figure 4G) when compared to TF.

### 2.6. Transcriptional Signature of Triploid Kidney ghrh-gh-sst-igf Axis in Response to Seawater Challenge

Overall transcriptional profiles of kidney *ghrh-gh-sst-igf* axis were shown by a heatmap (Figure 5A). The PCA showed distinct discrimination of *ghrh-gh-sst-igf* transcriptional signature between DS and TS (Figure 5B). Genes of *sst1b*, *sst2*, *sst5*, *sstr1b1*, *sstr5a*, *sstr5b*, *igfbp1a1*, *igfbp2b1*, *igfbp5a1*, *igfbp5a2*, and *igfbp6a1* resulted in transcriptional discrimination between DS and TS (Figure 5C). The PLS-DA exhibited distinct discriminations among groups of DS vs. TS and groups of TS vs. TF (Figure 5D). The loading plot of PLS-DA showed the transcriptional discriminations resulted from the genes of *sst1b*, *sst2*, *sst5*, *sstr1b1*, *sstr5a*, *sstr5b*, *igfbp2b1*, *igfbp5a1*, *igfbp5a2*, and *igfbp6a1* (Figure 5E). Compared to DS, the TS group exhibited up-regulated *sst1b*, *sstr1b1*, *sstr5a*, *sstr5b*, *igfbp5a1*, *igfbp5a2*, and *igfbp6a1* and decreased *ghrhrl1*, *sst2*, *igf1a2*, *igfbp5b1*, *igfbp5b2*, and *igfbp6b2* (Figure 5F). Compared to TF, trout of TS exerted up-regulated *sstr5a*, *sstr5b*, *igfbp1a1*, *igfbp5a1*, *igfbp5a2*, and *igfbp6a1* and decreased *sst1b*, *sst2*, *sst5*, *igf1a1*, *igf1a2*, *igfbp4a*, *igfbp5b2*, and *igfbp6b2* (Figure 5G).

### 2.7. Correlation Analyses of ghrh-gh-sst-igf Axis

The overall transcriptional correlations of the *ghrh-gh-sst-igf* axis in brain, kidney, and liver were analyzed via hierarchical clustering heatmaps (Figure 6A,G,L). In brain, positive correlations were observed between *ghra1* and *sstr1a1*, *ghra1* and *sstr2a2*, *sstr1a1* and *sstr3a*, *sstr2a2* and *sstr2b2*, *igf1b/igf3* and *igfbp5a2* (Figure 5B–F). In kidney, *igf1a2* exerted positive correlations with *igfbp5b2* and *igfbp6b2* (Figure 6H,I). The kidney *igfbp2a1* exerted positive correlations with *igfbp2a2*, and kidney *igfbp5b2* exhibited positive correlations with *igfbp6b2* (Figure 6J,K). In liver, the data showed negative correlations between *ghra2* and *igf1a1*, *ghra2* and *igf1a2*, *sstr2b1* and *igfbp6a2*, *igf1a2* and *igfbp5b2*, and *igfbp1b2* and *igfbp2a2*, while positive correlations between *ghra2* and *igfbp1b2*, *sstr2b1* and *sstr2b2*, *igf1a1* and *igf1a2*, *igfbp2a1* and *igfbp2a2*, and *igfbp2a1* and *igfbp5a1* (Figure 6M–V).

## 3. Discussion

### 3.1. Complete Repertoire of sst in Rainbow Trout

Our study identified seven *sst* genes of *sst1a*, *sst1b*, *sst2*, *sst3a*, *sst3b*, *sst5, sst6*. Similarly, six *sst* genes were identified in zebrafish with *sst1*–*sst6*, and *sst1*–*sst5* being observed in other teleosts of stickleback (*Gasterosteus aculeatus*), Japanese medaka (*Oryzias latipes*), puffer (*Takifugu rubripes*) and spotted green pufferfish (*Tetraodon nigroviridis*) [34]. The *sst1*, *sst2*, and *sst5* were generated due to two rounds of genome duplications (2R) in early vertebrate evolution [34,35], therefore we observed conserved *sst* members of *sst1*, *sst2*, and *sst5* in both salmonid and non-salmonid species. Previous studies indicated *sst3* and *sst6* were produced from tandem duplication of *sst1* and *sst2* [34,36]. Consistently, we observed *sst* subtypes were colocalized in chromosome 9 (*sst2*, *sst6*), chromosome 15 (*sst1b*, *sst3a*), or chromosome 27 (*sst1a*, *sst3b*), suggesting they were arranged in tandem repeat. The *sst4* is derived from *sst1* due to teleost specific genome duplications (or three rounds of genome duplications, 3R) [34]. However, the *sst4* was absent in rainbow trout. The salmonid ancestors experienced an additional round of genome duplication [37], and salmonid specific genome duplications (or 4R) probably result in *sst4* being lost and generate duplicated *sst1* and *sst3* duplications. Future studies should be focused on physiology and pharmacology between SST1 and SST4, and between SST1(4) paralogs.

### 3.2. Transcriptional Signature of ghrh-sst-gh-igf Axis in Response to Seawater Challenge

#### 3.2.1. The SST System

The SST exhibits a conserved function of suppressing growth in both mammals and teleosts [38,39,40,41]. Triploid trout showed significantly down-regulated brain *sstr1b1* and kidney *sst2*, potentially suggesting triploid trout exerts a better growth after seawater acclimation. A recent human clinical case report showed SST and analogs target the GI tract and pancreas, thus regulating water and salt reabsorption [42]. We observed gene expressions of *sst* (*sst1b*, *sst2*, *sst5*) and *sstr* (*sstr1b1*, *sstr5a*, *sstr5b*) were changed in kidney rather than liver after seawater challenge. Considering the pleiotropic physiology of GH in regulating growth and smoltification, we proposed that SST might regulate extracellular osmolarity via the GH-IGF axis in kidney [15,43]. Consistently, recent studies showed SST is involved in physiological regulations of GHRs, IGFs, IGFRs, and IGFBPs [44,45,46].

#### 3.2.2. The IGFBPs System

Smolting is a pan-hyperendocrine state in salmonids, and McCormick indicated that the GH-IGF axis is significantly involved in migratory readiness via regulating salinity tolerance, growth, and metabolism. IGF acts as the primary regulator and target of GH functions, and the local physiology of IGF is co-regulated by IGFRs and IGFBPs. Salmonid specific genome duplication results in expanded *igfbp* paralogs, however, physiology of these novel *igfbp* paralogs in response to seawater challenge (migratory readiness) has yet to be determined.

The IGFBPs of triploid trout before and after seawater challenge.

In triploid trout, the upregulated brain *igfbp1a2* and kidney *igfbp1a1* were observed after seawater challenge, suggesting the central and peripheral endocrine response to salinity challenge is regulated by different *igfbp1a* subtypes. Previous studies showed the upregulated IGFBP1 could suppress the interaction between IGF and receptors, thus saving the energy from basal metabolism for the processes of acclimation to environmental stress [47,48]. Meanwhile, triploid trout kidney showed an increased trend of *igfbp5a* subtype expression and significantly downregulated *igfbp5b2* and *igfbp6b2*. The IGFBP5 and IGFBP6 subtypes are involved in regulation of ionic and energy (nutrition) homeostasis. For example, *igfbp5a* is specifically expressed in zebrafish ionocytes (chloride cell) and dysregulation of IGFBP5a results in abnormal ionocyte proliferation [49]. Consistent to previous studies showing *igfbp5a* and *igfbp5b* exert distinctive expressions in zebrafish and grass carp, we showed *igfbp5(6)a* subtypes exhibited conversed transcriptional signatures with *igfbp5(6)b*, suggesting an evolutionary divergence in physiology [50,51].

2.Comparison of IGFBPs in diploid and triploid in response to seawater challenge.

It is necessary to evaluate whether triploid trout exhibit equally biology and physiology performances as diploids in response to environment challenges before mass commercial production [52]. The GH-IGF axis is involved in multiple functions associated with commercially relevant traits, including growth, immunity, and smoltification. IGFBP1 could be induced by stress and acts as a negative regulator of teleost growth and development [24]. Diploid showed up-regulated brain *igfbp1a1* (*p =* 0.057) after seawater challenge. Brain plays a key role in stress reactivity and modulates the physiological and behavioral alterations for stress coping and recovery [53]. Meanwhile, significantly increased brain *igfbp2a1* and *igfbp2a2* expressions were observed in diploid trout. Previous study showed *igfbp2* overexpression results in growth and development reduction [54,55]. Our results suggested triploid were less sensitive to seawater challenge when compared to diploids. The *igfbp5* exerted subtype-specific transcription patterns in kidney (down-regulated *igfbp5b2*) and liver (down-regulated *igfbp5a2*) in triploid when compared to diploid. Previous studies revealed *igfbp5* is involved in both muscle growth and ionic homeostasis in teleost [24,27,49,56,57,58]. The triploid exhibited potential IGFBP5 subtype-regulated crosstalk between growth (or energy production) and osmoregulation in liver and kidney. The *igfbp6*, a growth inhibitor of teleost growth [24], showed down-regulation in triploid after seawater challenge, suggesting a better energy crosstalk between seawater acclimation and growth in triploid trout.

## 4. Materials and Methods

### 4.1. Ethics Statement

All experimental protocols were performed in conformity to the Guidelines of Animal Research and Ethics Committees of the Ocean University of China (Permit Number: 20141201), and National Institutes of Health Guide for the Care and Use of Laboratory Animals (NIH Publications, No. 8023, revised 1978). Immature trout were used in this study and sex effects were not considered. No endangered or protected animals were used in this study.

### 4.2. Ploidy Identification

The ploidy identification procedure was combined with CyFlow^TM^Robby6 (Sysmex, Norderstedt, Germany). Briefly, caudal fin (~0.05 g) was clipped and homogenized with 1×PBS and CyStain^TM^ UV Precise P Nuclei Extraction Buffer (Sysmex, Germany). At least 10,000 cells were measured per sample via flow cytometry. The average relative DNA content of six diploid trout were determined as the diploid control.

### 4.3. Animals Acclimation and Salinity Challenge

Trout juveniles (diploid and triploid, ~9.7 g, ~10.7 cm) were obtained from Linqu Salmon and Trout Aquatic Breeding LLC (Weifang, Shandong, China). The experimental trout were derived from the same full-sibling family batch with same day age and synchronized development. Trout were acclimated for 10 days before salinity challenge. Fish were acclimated and maintained in a recirculating water system at the Experimental Fish Facility in the Key Laboratory of Mariculture, Ocean University of China, with ~16 °C of water temperature, ~7 mg/L of dissolved oxygen, saturated feeding (5% of body weight), and natural photoperiod of 12:12 (hours) of light and dark. The acclimation protocols were performed in conformity with the Standards of Linxia Salmon and National Trout Elite Breeding and Protection Farm (Linxia, Gansu, China, Approved by the Department of Agriculture, China, 2009).

After acclimation, trout were divided into four groups (diploid freshwater exposure (DF), diploid seawater exposure (DS), triploid freshwater exposure (TF), and triploid seawater exposure (TS)). Each group contained three biological replications and each replication contained 18 individual trout. The DF and TF groups were cultured in cuboid tanks filled with freshwater, while the DS and TS groups were cultured in cuboid tanks filled with seawater at salinity of 15 (15 ppt) for 7 days. Trout were reared in the same recirculating water system with acclimation protocol, with the same water temperature, dissolved oxygen, feeding regime, and photoperiod. The exposure seawater (15 ppt) was prepared by dilution of the seawater (30 ppt) with aerated freshwater, and the salinity for the exposure media was detected by LS10T (Suwei, Guangdong, China). During exposure, 40% of the exposure media was replaced every two days for the following reasons: (1) to maintain the salinity concentrations, (2) to qualify the water quality, (3) to reduce the animal chemicals. No mortality was observed in DS and TS groups after a 24-h salinity exposure, and the survival rates on day 7 in DF, DS, TF, and TS were 100%, 96.3 ± 3.2%, 98.1 ± 3.2%, and 98.1 ± 3.2%, respectively.

### 4.4. Sampling and RNA-seq

Trout were starved 24 h prior to sacrifice and then anesthetized with tricaine (MS-222, Sigma-Aldrich, Shanghai, China). Tissues of brain, liver, and kidney were quickly removed and then stored in tubes at −80 °C in RNase-free conditions. Total RNA was extracted from brain, liver, and kidney via TRIzol reagent (Vazyme Biotech, Nanjing, China) based on the manufacturer’s protocol. The quality and quantity of extracted RNA were evaluated with agarose gel electrophoresis and a biophotometer (OSTC, Shanghai, China).

RNA sequences and data analyses were performed with the general protocol and pipeline for medical and fishery studies of the commercial provider (OEbiotech, Shanghai, China). To reduce the individual differences of RNA-Seq, one RNA-Seq library was constructed via pooling equal quantities of RNA from two individuals within the same replicate (tank). The libraries were sequenced with Illumina HiSeq 4000 platform via a commercial provider (Oebiotech, Shanghai, China). The clean reads of sequences were derived after removing raw reads of low quality and/or poly-N were removed. The latest rainbow trout genome (GCA_013265735.3) was used for mapping of the clean reads via histat2 [59]. The DESeq2 R package was used for gene expression analyses [60], and normalized count (via DESeq2 R package) was used in this study. Details of the data qualities and expression metrics are shown in Appendix A. Sequence reads are available in the Sequence Read Archive Database (PRJNA844477).

### 4.5. Identification of Rainbow Trout sst Repertoire

Rainbow trout genomic data were used to identify *sst* repertoire. The SST amino acid sequences of zebrafish (*Danio rerio*) and goldfish (*Carassius auratus*) were used for sequence comparisons via TBLASTN. Molecular weight (MW), isoelectric points (pI), and chromosome location of SST genes/proteins were predicted/acquired by online resources of ProtParam tool and NCBI.

Phylogenetic tree was conducted with the amino acid sequences of SSTs from rainbow trout (*Oncorhynchus mykiss*), human (*Homo sapiens*), house mouse (*Mus musculus*), chicken (*Gallus gallus*), zebrafish (*Danio rerio*), Japanese medaka (*Oryzias latipes*), goldfish (*Carassius auratus*), common carp (*Cyprinus carpio*), fugu (*Takifugu rubripes*), channel catfish (*Ictalurus punctatus*), and Atlantic salmon (*Salmo salar*). ClustalW in MEGA 11.0 was used for protein sequence alignment [61]. Phylogenetic tree was conducted via the neighbor-joining (NJ) method and Jones–Taylor–Thornton (JTT) model in MEGA 11.0 software with 1000 bootstrap replicates [62]. The chromosome localization of *sst* genes in rainbow trout were visualized by TBtools. The *sst* gene information of rainbow trout, Atlantic salmon, pink salmon (*Oncorhynchus gorbuscha*), zebrafish, and Japanese medaka were used for syntenic analysis. The *sst* gene structures of rainbow trout were visualized by online tools (http://gsds.gao-lab.org/index.php, accessed on 27 March 2022). The SST motif analysis was conducted via MEME website (https://meme-suite.org/meme/tools/meme, accessed on 27 March 2022). Multiple sequence alignments among species were retrieved from the ESPript 3.0 website (https://espript.ibcp.fr/ESPript/ESPript/index.php, accessed on 3 April 2022).

### 4.6. Quantitative Polymerase Chain Reaction (qPCR)

Results of RNA-Seq were validated via qPCR (Appendix A). Total RNA was reverse-transcribed into cDNA using PrimeScript RT reagent kit (Takara, Shiga, Japan). The 4× diluted cDNA was served as the template for qPCR. The primers were designed via Primer 5 software (Appendix A) and *β-actin* was selected as the reference gene [63]. The qPCR system was a 10 µL reaction, containing 1 µL cDNA, 5 µL SYBR^®^FAST qPCR Master Mix (Monad, Wuhan, China), 0.2 µL of each primer, and 3.6 µL RNAase-free water. The reaction program was 95 °C for 30 s, 40 cycles of 95 °C for 10 s, and Tm for 30 s, followed by 72 °C for 30 s. The qPCR was carried out via StepOnePlus™ Real-time PCR system (Applied Biosystems, Foster City, CA, USA). The results were calculated by the 2^−^^ΔΔCt^ method [64].

### 4.7. Data Visualization and Statistical Analysis

According to previous studies in medical and fishery studies [65,66], the count of RNA-Seq data were normalized by DESeq2 [67]. The normalized data were analyzed via the websites of MetaboAnalyst and NetworkAnalyst for multivariate analysis of principal components analysis (PCA), partial least squares discriminant analysis, loading plots, and clustering algorithm (heatmap) [68]. Genes were selected via the screening criteria of Log2|Foldchange| ≥ 1 or *p* value < 0.05 (Log_10_(count+1) tested by Student’s *t*-test). The correlation analysis of gene expressions was analyzed via the Pearson correlation coefficient by MetaboAnalyst and GraphPad Prism 8.0. Data of basal expression of *ghrh-gh-sst-igf* axis in diploid trout were retrieved from previously published RNA-Seq data (SRA of NCBI: PRJNA865462 for liver; SRA of NCBI: PRJNA753277 for brain and kidney) [69,70]. The gene count data were analyzed by DESeq2 R package and then the average count was normalized by average *β-actin* ((average of normalized count × 10,000)/(average of *β-actin* count)) [60]. Expression matrix are shown in Appendix A.

## 5. Conclusions

In this study, we showed that salmonids exhibited increased *sst* gene duplications due to salmonid specific genome duplication. Similar to GH, the SST and SSTR exhibited pleiotropic effects in modulating diverse physiological processes including growth and smoltification. Compared to diploid, triploid trout exhibited a better seawater acclimation. The *igfbp5* and *igfbp6* subtypes were involved in energy redistribution between growth and osmoregulatory regulation.

## Figures and Tables

**Figure 1 ijms-23-08691-f001:**
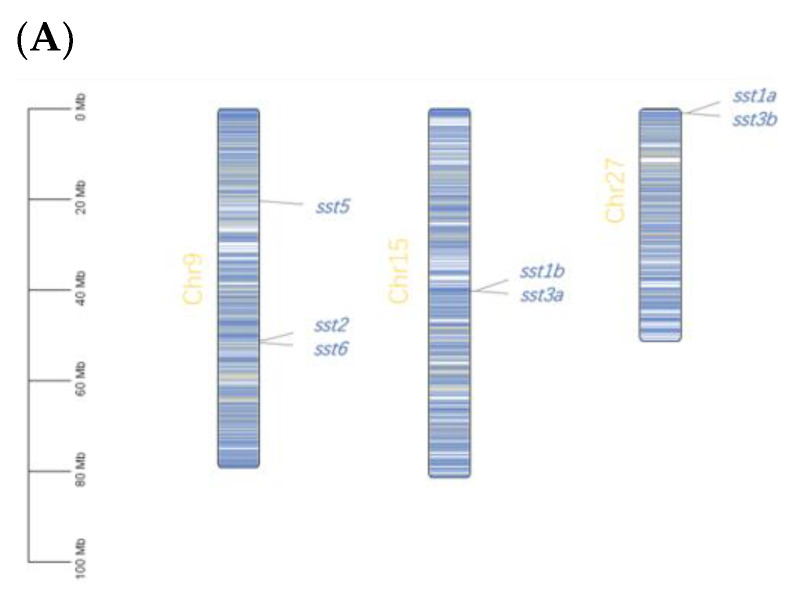
Chromosomal distribution, phylogenetic, syntenic analysis of *sst* genes in rainbow trout. (**A**) The chromosomal localization of *sst* genes. (**B**) Phylogenetic tree of the SST family in vertebrate. The amino acid sequences were analyzed via a neighbor-joining method with 1000 bootstrap replications. Rainbow trout SSTs are highlighted by a red pentacle. (**C**) Gene structure and protein motif of SST proteins of rainbow trout. (I) Exon and intron structures of SST proteins in rainbow trout. Phylogenetic tree was constructed by a neighbor-joining method with 1000 bootstrap replications. (II) Protein motif analyses of SST proteins in rainbow trout. Eight specific motifs were found in SST proteins of rainbow trout, different colors represent different motifs. (**D**–**G**) Conserved syntenic analyses of *sst* genes: *sst1a*/*b* and *sst3a*/*b* (1D); *sst2* (1E); *sst5* (1F); *sst6* (1G). *sst* genes are shown by the same color. The double slash represents the omission of more than two genes.

**Figure 2 ijms-23-08691-f002:**
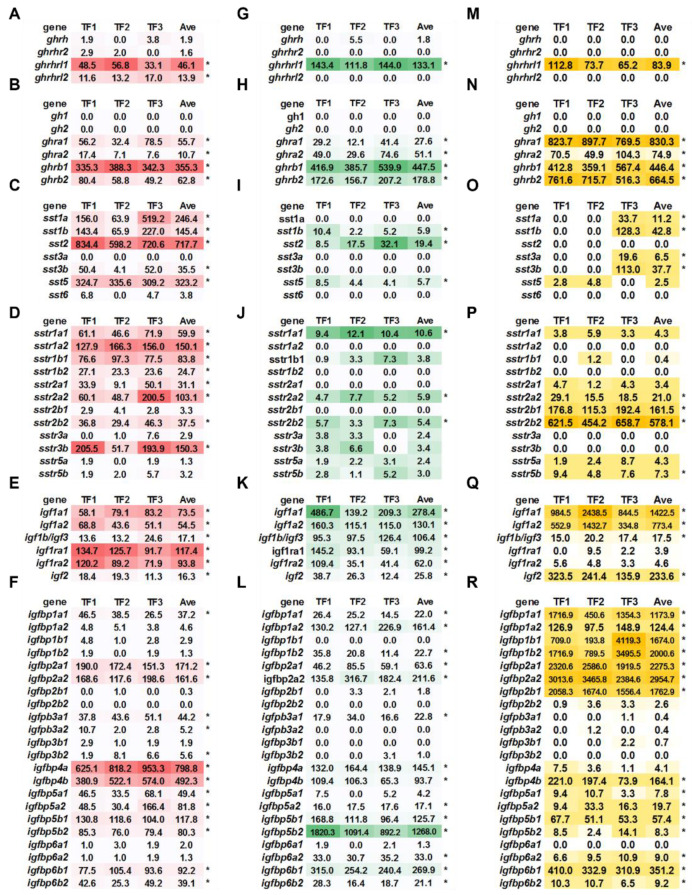
The basal expression of *ghrh-sst-gh-igf* axis in triploid trout. (**A**–**F**) The gene expression of *ghrh* and receptors (**A**), *gh* and receptors (**B**), *sst* (**C**), *sstr* (**D**), *igf* and receptors (**E**), and *igfbps* (**F**) in brain. (**G**–**L**) The gene expression of *ghrh* and receptors (**G**), *gh* and receptors (**H**), *sst* (**I**), *sstr* (**J**), *igf* and receptors (**K**), and *igfbps* (**L**) in kidney. (**M**–**R**) The gene expression of *ghrh* and receptors (**M**), *gh* and receptors (**N**), *sst* (**O**), *sstr* (**P**), *igf* and receptors (**Q**), and *igfbps* (**R**) in liver. The gene expression levels were generated via counts normalized by DESeq2. The “*” indicates average count > 5.

**Figure 3 ijms-23-08691-f003:**
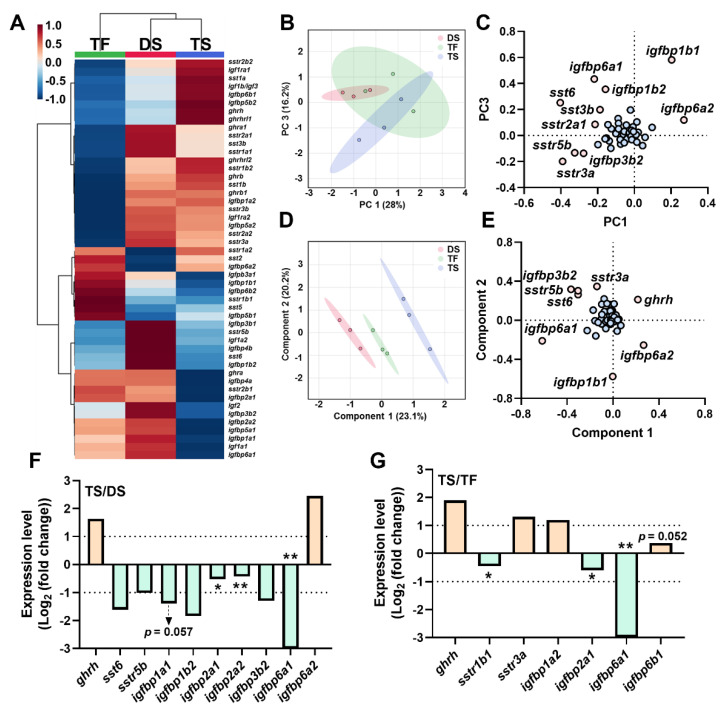
The transcriptional signature of *ghrh-sst-gh-igf* axis in brain in diploid and triploid trout. (**A**) The heatmap of *ghrh-sst-gh-igf* axis in brain. (**B**,**C**) Principal component analysis (**B**) and loading plot (**C**) of *ghrh-sst-gh-igf* axis. (**D**,**E**) Partial least squares discriminant analysis (**D**) and loading plot (**E**) of *ghrh-sst-gh-igf* axis. In loading plot, gene(s) further away from center point (0, 0) showed obvious effects on principal component analysis/partial least squares-discriminant analysis. (**F**,**G**) Key genes of *ghrh-sst-gh-igf* axis of TS/DS (**F**) and TS/TF (**G**). Genes were selected based on the principle of Log_2_|Foldchange| ≥ 1 or *p* value < 0.05 (Log_10_(count+1) by Student’s *t*-test, “*” *p* < 0.05; “**” *p* < 0.01). TS: **t**riploid **s**eawater exposure; DS: **d**iploid **s**eawater exposure group; TF: **t**riploid **f**reshwater exposure group.

**Figure 4 ijms-23-08691-f004:**
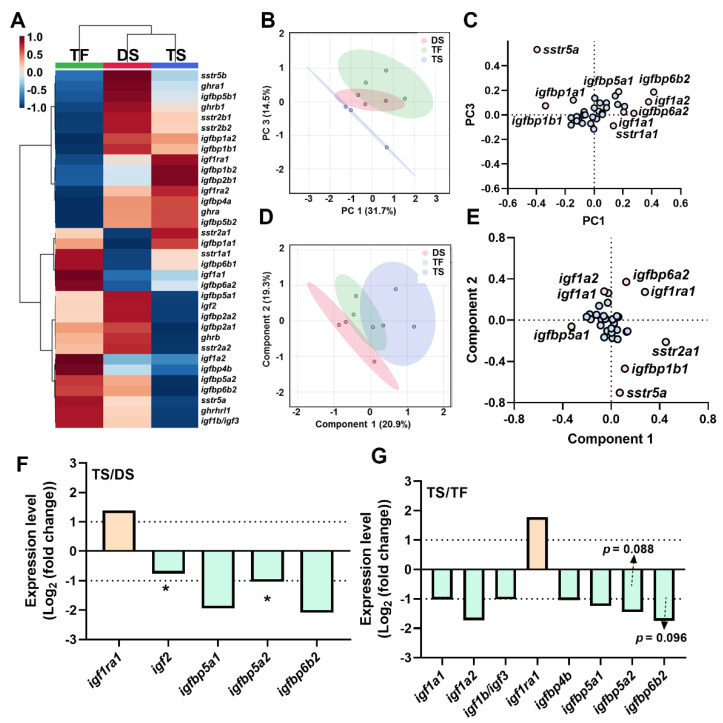
The transcriptional signature of *ghrh-sst-gh-igf* axis in liver in diploid and triploid trout. (**A**) The heatmap of *ghrh-sst-gh-igf* axis in brain. (**B**,**C**) Principal component analysis (**B**) and loading plot (**C**) of *ghrh-sst-gh-igf* axis. (**D**,**E**) Partial least squares discriminant analysis (**D**) and loading plot (**E**) of *ghrh-sst-gh-igf* axis. In loading plot, gene(s) further away from center point (0, 0) showed obvious effects on principal component analysis/partial least squares-discriminant analysis. (**F**,**G**) Key genes of *ghrh-sst-gh-igf* axis of TS/DS (**F**) and TS/TF(**G**). Genes were selected based on the principle of Log_2_|Foldchange| ≥ 1 or *p* value < 0.05 (Log_10_(count+1) by Student’s *t*-test, “*” *p* < 0.05). TS: **t**riploid **s**eawater exposure; DS: **d**iploid **s**eawater exposure group; TF: **t**riploid **f**reshwater exposure group.

**Figure 5 ijms-23-08691-f005:**
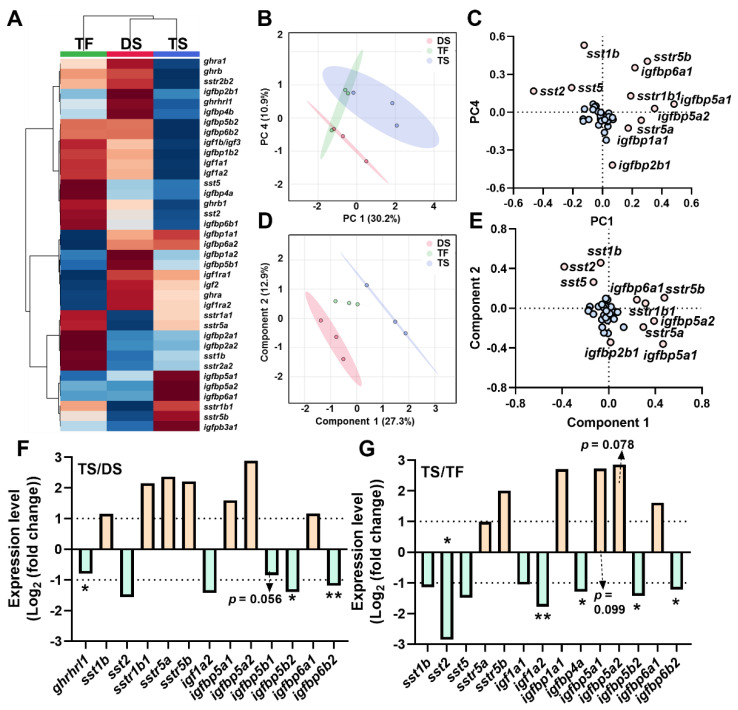
The transcriptional signature of *ghrh-sst-gh-igf* axis in kidney in diploid and triploid trout. (**A**) The heatmap of *ghrh-sst-gh-igf* axis in brain. (**B**,**C**) Principal component analysis (**B**) and loading plot (**C**) of *ghrh-sst-gh-igf* axis. D and E: Partial least squares discriminant analysis (**D**) and loading plot (**E**) of *ghrh-sst-gh-igf* axis. In loading plot, gene(s) further away from center point (0, 0) showed obvious effects on principal component analysis/partial least squares-discriminant analysis. (**F**,**G**) Key genes of *ghrh-sst-gh-igf* axis of TS/DS (**F**) and TS/TF (**G**). Genes were selected based on the principle of Log_2_|Foldchange| ≥ 1 or *p* value < 0.05 (Log_10_(count+1) by Student’s *t*-test, “*” *p* < 0.05; “**” *p* < 0.01). TS: **t**riploid **s**eawater exposure; DS: **d**iploid **s**eawater exposure group; TF: **t**riploid **f**reshwater exposure group.

**Figure 6 ijms-23-08691-f006:**
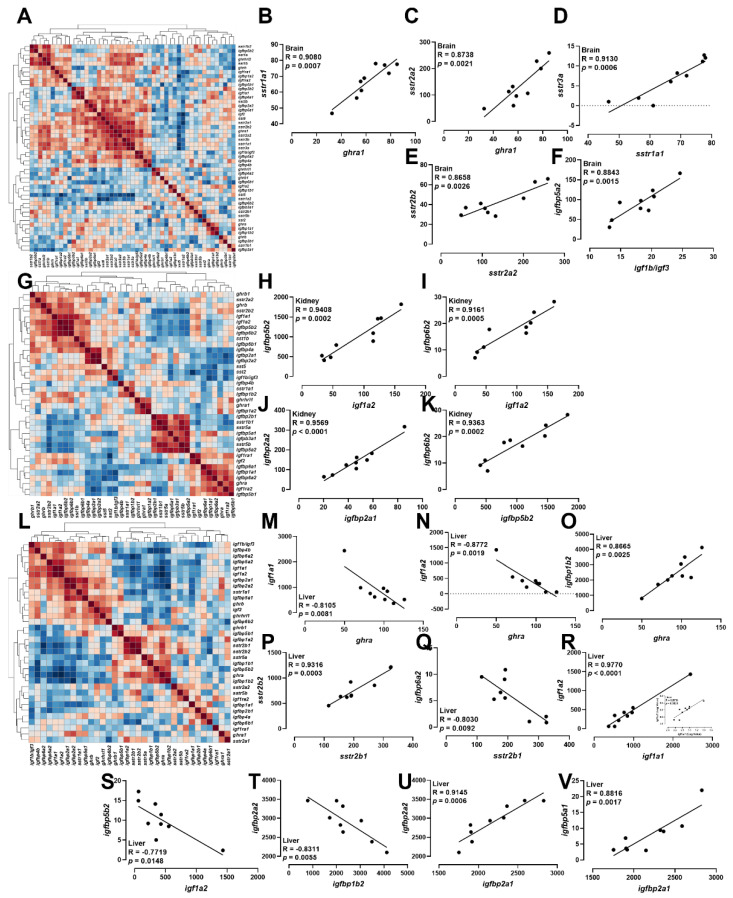
Correlation analysis of *ghrh-sst-gh-igf* axis in brain, kidney, and liver. (**A**–**F**) Heatmap of correlation of *ghrh-sst-gh-igf* axis in brain (**A**) and the Pearson correlation coefficient of two genes in brain (**B**–**F**). (**G**–**K**) Heatmap of correlation of *ghrh-sst-gh-igf* axis in kidney (**G**) and the Pearson correlation coefficient of two genes in kidney (**H**–**K**). (**L**–**V**) Heatmap of correlation of *ghrh-sst-gh-igf* axis in liver (**L**) and the Pearson correlation coefficient of two genes in liver (**M**–**V**).

**Table 1 ijms-23-08691-t001:** The sequence information and accession number of *sst* family members.

Gene Name	Chromosome ID	mRNA (bp)	Protein Length (aa)	MW (KDa)	pI	Accession Numbers
*sst1a*	27	771	114	12.42032	5.51	XP_021442876.1
*sst1b*	15	785	114	12.46135	5.14	XP_021418859.1
*sst2*	9	887	111	12.47253	6.31	XP_021471775.1
*sst3a*	15	624	115	12.96316	10.16	NP_001118175.1
*sst3b*	27	641	111	12.32508	7.66	XP_021442875.1
*sst5*	9	3822	108	12.06404	6.10	XP_021470745.1
*sst6*	9	943	98	11.50144	8.54	XP_021472825.1

**Table 2 ijms-23-08691-t002:** Comparison of *ghrh-gh-sst-igf* axis between diploid and triploid trout.

Brain				Liver				Kidney			
Gene	Triploid	Diploid	Fold	Gene	Triploid	Diploid	Fold	Gene	Triploid	Diploid	Fold
*ghrh*	2.19	0.57	3.82	*ghrb1*	2761.29	6143.88	0.45	*ghrh*	1.04	0.11	9.17
*ghrhr2*	1.89	0.61	3.11	*sst1b*	264.44	116.81	2.26	*ghrhrl2*	0.00	0.73	0.00
*sst1a*	282.66	60.84	4.65	*sst1a*	69.47	124.07	0.56	*ghrhrl1*	75.59	21.00	3.60
*sst3b*	40.72	5.74	7.10	*sst3b*	233.07	61.20	3.81	*ghra1*	15.67	7.07	2.22
*sst5*	370.77	69.89	5.30	*sstr1a1*	26.74	8.78	3.05	*ghrb1*	254.22	133.56	1.90
*sstr2a2*	118.27	48.68	2.43	*igf1a1*	8799.16	2417.77	3.64	*sst1b*	3.36	0.74	4.57
*sstr3b*	172.49	32.15	5.37	*igf1a2*	4784.27	1005.80	4.76	*sst5*	3.22	1.09	2.96
*sstr5a*	1.46	3.89	0.38	*igfbp1a1*	7261.62	13,604.07	0.53	*sstr1a1*	6.03	3.51	1.72
*igfbp1b1*	3.33	0.00		*igfbp3a2*	2.45	10.94	0.22	*sstr5b*	1.72	0.27	6.40
*igfbp3a2*	5.94	1.71	3.46	*igfbp4a*	25.11	44.17	0.57	*igf1a1*	158.15	29.80	5.31
*igfbp3b2*	6.37	13.75	0.46	*igfbp4b*	1014.94	231.54	4.38	*igf1a2*	73.93	9.65	7.66
*igfbp5a2*	93.82	24.07	3.90	*igfbp5a1*	48.17	178.17	0.27	*igf1ra1*	56.33	23.54	2.39
*igfbp6a1*	2.26	4.07	0.55	*igfbp6a1*	0.00	0.78	0.00	*igfbp1a2*	91.69	30.75	2.98
								*igfbp2a2*	120.22	241.84	0.50
								*igfbp3a2*	0.00	0.49	0.00
								*igfbp4a*	82.43	40.55	2.03
								*igfbp6b1*	153.32	83.79	1.83

## Data Availability

Data are contained within the article.

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
