# Peer review of "Crosstalk between Growth and Osmoregulation of GHRH-SST-GH-IGF Axis in Triploid Rainbow Trout (Oncorhynchus mykiss)"

_ijms, 2022, doi:10.3390/ijms23158691_

Round 1
Reviewer 1 Report
MDPI
International Journal of Molecular Sciences
MDPI-IJMS-1829016
In this manuscript, the authors studied the ghrh-sst-gh-igf axis of diploid and triploid in response to seawater challenge. The authors concluded and reported that CHRH-SST-GH-IGF axis exhibited pleiotropic effects in regulating growth and osmoregulatory regulation during trout which might provide insights into seawater aquaculture of salmonid species. The authors have provided novel information in the growth and osmoregulation of GHRH-SST-GH-IGF axis in triploid rainbow trout.
Author Response
Dear reviewer #1:
We would like to thank you for your revision and positive comments, which provides great encouragements to us. We have made some revisions (according to Reviewer #2) to the manuscript.
Sincerely,
Hai-Shen Wen and Kai-Wen Xiang
Reviewer 2 Report
The deals with an interesting topic and increase our knowledge about the growth and osmoregulation of GHRH-SST-GH-IGF axis in triploid rainbow trout. the ms can be considered after minor modifications :
Abstract
results must be written with statistical view
Keywords suggested to be changed
Introduction
Line 32-36 no reference provided
The last part of the introduction must clearly the aim
Author Response
Dear reviewer#2:
Thank you for your comments on our manuscript. These comments are all valuable and very helpful for revising and improving our paper, as well as the important guiding significance to our research. We have studied comments carefully and the responds to the comments are as follows point by point.
Suggestion 1: Results must be written with statistical view in the Abstract.
Answer: The statistical view of results was added into the Abstract (as well as in Figures 3 - 5).
Please see the revisited content in line 19-32.
The statistical view of figures was shown in figure legends of Figures 3 - 5.
Suggestion 2: Keywords suggested to be changed.
Answer: Keywords were replaced by triploid rainbow trout; ghrh-gh-sst-igf axis; osmoregulation.
Suggestion 3: Line 32-36 no reference provided.
Answer: We cited two references work on the rainbow trout and steelhead trout.
- Pavlov, D.; Savvaitova, K. On the problem of ratio of anadromy and residence in salmonids (Salmonidae). J. Ichthyol. 2008, 48, 778-791.
- Kendall, N.W.; McMillan, J.R.; Sloat, M.R.; Buehrens, T.W.; Quinn, T.P.; Pess, G.R.; Kuzishchin, K.V.; McClure, M.M.; Zabel, R.W. Anadromy and residency in steelhead and rainbow trout (Oncorhynchus mykiss): a review of the processes and patterns. Can. J. Fish. Aquat. Sci. 2015, 72, 319-342, https://doi.org/10.1139/cjfas-2014-0192.
Please see the revisited content in line 44.
Suggestion 4: The last part of the introduction must clearly the aim.
Answer: The last part of the introduction has been revised to highlight the aim of this article:
“To further understand the effect of GHRH-SST-GH-IGF axis in seawater exposure be-tween diploid and triploid trout and explore the novel regulatory mechanisms of trout osmoregulation and the candidates of genetic breeding for seawater trout culture.”
Please see the revisited content in line 96-104.
Although the editors and reviewers do not ask to improve the discussion, we also improved the discussion part. We rescheduled with new references, thus providing a more comprehensive and clear discussion of GHRH-SST-GH-IGF axis in osmoregulatory regulation.
We have already resubmitted the revised manuscript. In the revised manuscript, changes according to your comments were highlighted by “Track Changes” function. Everything is subject to our latest uploaded manuscript.
Sincerely,
Hai-Shen Wen and Kai-Wen Xiang